# Learning to Optimize for Reinforcement Learning

## Abstract

In recent years, by leveraging more data, computation, and diverse tasks, learned optimizers have achieved remarkable success in supervised learning, outperforming classical hand-designed optimizers. Reinforcement learning (RL) is essentially different from supervised learning and in practice these learned optimizers do not work well even in simple RL tasks. We investigate this phenomenon and identity three issues. First, the gradients of an RL agent vary across a wide range in logarithms while their absolute values are in a small range, making neural networks hard to obtain accurate parameter updates. Second, the agent-gradient distribution is non-independent and identically distributed, leading to inefficient meta-training. Finally, due to highly stochastic agent-environment interactions, the agent-gradients have high bias and variance, which increase the difficulty of learning an optimizer for RL. We propose gradient processing, pipeline training, and a novel optimizer structure with good inductive bias to address these issues. By applying these techniques, for the first time, we show that learning an optimizer for RL from scratch is possible. Although only trained in toy tasks, our learned optimizer can generalize to unseen complex tasks in Brax.

## 1 Introduction

Deep learning has achieved great success in many areas (LeCun et al. 2015), which is largely attributed to the automatically learned features that surpass handcrafted expert features. The use of gradient descent enables automatic adjustments of parameters within a model, yielding highly effective features. Despite these advancements, as another important component in deep learning, optimizers are still largely hand-designed and heavily reliant on expert knowledge. To reduce the burden of hand-designing optimizers, researchers propose to learn to optimize with the help of meta-learning (Sutton 1992, Andrychowicz et al. 2016, Chen et al. 2017, Wichrowska et al. 2017, Maheswaranathan et al. 2021). Compared to designing optimizers with human expert knowledge, learning an optimizer is a data-driven approach, reducing the reliance on expert knowledge. During training, a learned optimizer can be optimized to speed learning and help achieve better performance.

Despite the significant progress in learning optimizers, previous works only present learned optimizers for supervised learning (SL). These learned optimizers usually have complex neural network structures and incorporate numerous human-designed input features, requiring a large amount of computation and human effort to design and train them. Moreover, they have been shown to perform poorly in reinforcement learning (RL) tasks (Metz et al. 2020b; 2022b). *Learning to optimize for RL remains an open and challenging problem.*

Classical optimizers are typically designed for optimization in SL tasks and then applied to RL tasks. However, RL tasks possess unique properties largely overlooked by classical optimizers. For example, unlike SL, the input distribution of an RL agent is non-stationary and non-independent and identically distributed (non-iid) due to locally correlated transition dynamics (Alt et al. 2019). Additionally, due to policy and value iterations, the target function and the loss landscapes in RL are constantly changing throughout the learning process, resulting in a much more unstable and complex optimization process. In some cases, these properties also make it inappropriate to apply optimization algorithms designed for SL to RL directly, such as stale accumulated gradients (Bengio et al. 2020a) or unique interference-generalization phenomenon (Bengio et al. 2020b). *We still lack optimizers specifically designed for RL tasks.*

In this work, we aim to learn optimizers for RL. Instead of manually designing optimizers by studying RL optimization, we apply meta-learning to learn optimizers from data generated in the agent-environment interactions. We first investigate the failure of traditional learned optimizers in RL tasks and find that the complicated agent-gradient distribution impedes the training of learned optimizers for RL. Specifically, it is difficult to obtain accurate parameter updates due to the logarithmic variation of agent-gradients across a wide range, despite their small absolute values. Furthermore, the non-iid nature of the agent-gradient distribution also hinders meta-training. Lastly, the highly stochastic agent-environment interactions can lead to agent-gradients with high bias and variance, exacerbating the difficulty of learning an optimizer for RL. In response to these challenges, we propose a novel approach, *Optim4RL*, a learned optimizer for RL that involves gradient processing, pipeline training, and a specialized optimizer structure with good inductive bias. Compared with previous works, Optim4RL is more stable to train and more effective in optimizing RL tasks, without complex optimizer structures or numerous human-designed input features. We demonstrate that Optim4RL can learn to optimize RL tasks from scratch and generalize to unseen tasks. Our work is the first to propose a learned optimizer for deep RL tasks that works well in practice.

## 2 BACKGROUND

### 2.1 REINFORCEMENT LEARNING

The process of reinforcement learning (RL) can be formalized as a Markov decision process (MDP). Formally, let $M = (\mathcal{S}, \mathcal{A}, \mathrm{P}, r, \gamma)$ be an MDP which includes a state space $\mathcal{S}$, an action space $\mathcal{A}$, a state transition probability function $\mathrm{P} : \mathcal{S} \times \mathcal{A} \times \mathcal{S} \to \mathbb{R}$, a reward function $r : \mathcal{S} \times \mathcal{A} \to \mathbb{R}$, and a discount factor $\gamma \in [0, 1)$. At each time-step $t$, the agent observes a state $S_t \in \mathcal{S}$ and samples an action $A_t$ by the policy $\pi(\cdot|S_t)$. Then it observes the next state $S_{t+1} \in \mathcal{S}$ according to $\mathrm{P}$ and receives a scalar reward $R_{t+1} = r(S_t, A_t)$. The return is defined as the weighted sum of rewards, i.e., $G_t = \sum_{k=t}^{\infty} \gamma^{k-t} R_{k+1}$. The state-value function $v_\pi(s)$ is defined as the expected return starting from a state $s$. The agent aims to find an optimal policy $\pi^*$ to maximize the expected return.

Proximal policy optimization (PPO) (Schulman et al. 2017) and advantage actor-critic (A2C) (Mnih et al. 2016) are two widely used RL algorithms for continuous control. PPO improves training stability by using a clipped surrogate objective to prevent the policy from changing too much at each time step. A2C is a variant of actor-critic method (Sutton & Barto 2018), featured with multiple-actor parallel training and synchronized gradient update. In both algorithms, $v_\pi$ is approximated with $v$ which is usually parameterized as a neural network. In practice, temporal difference (TD) learning is applied to approximate $v_\pi$:

$$v(S_t) \leftarrow v(S_t) + \alpha(R_{t+1} + \gamma v(S_{t+1}) - v(S_t)), \tag{1}$$

where $\alpha$ is the learning rate, $S_t$ and $S_{t+1}$ are two successive states, and $R_{t+1} + \gamma v(S_{t+1})$ is named the TD target. TD targets are usually biased, non-stationary, and noisy due to changing state-values, complex state transitions, and noisy reward signals (Schulman et al. 2016). They usually induce a changing loss landscape that evolves during training. As a result, the agent-gradients [1] usually have high bias and variance which can lead to sub-optimal performance or even a failure of convergence.

### 2.2 LEARNING TO OPTIMIZE WITH META-LEARNING

We aim to learn an optimizer using meta-learning. Let $\theta$ be the agent-parameters of an RL agent that we aim to optimize. A (learned) optimizer is defined as an update function $U$ which maps input gradients to parameter updates, implemented as a meta-network, parameterized by the meta-parameters $\phi$. Let $z$ be the input of this meta-network which may include gradients $g$, losses $L$, exponential moving average of gradients, etc. Let $h$ be an optimizer state which stores historical values. We can then compute agent-parameters updates $\Delta\theta$ and the updated agent-parameters $\theta'$:

$$\Delta\theta, h' = U_\phi(z, h) \text{ and } \theta' = \theta + \Delta\theta.$$

Note that all classical first-order optimizers can be written in this form with $\phi = \emptyset$. As an illustration, for RMSProp (Tieleman & Hinton 2012), let $z = g$; $h$ is used to store the average of squared

---

[1]In this work, we use the term agent-gradient to refer to gradients of all parameters in a learning agent, which may include policy gradient, gradient of value functions, gradient of other hyper-parameters (e.g., the temperature in SAC (Haarnoja et al. 2018)).

gradients. Then $U_{\text{RMSProp}}(g, h) = (-\frac{\alpha g}{\sqrt{h' + \epsilon}}, h')$, where $h' = \beta h + (1 - \beta)g^2$, $\beta \in [0, 1]$, and $\epsilon$ is a tiny positive number for numerical stability.

Similar to Xu et al. (2020), we apply bilevel optimization to optimize $\theta$ and $\phi$. First, we collect $M + 1$ trajectories $\mathcal{T} = \{\tau_i, \tau_{i+1}, \cdots, \tau_{i+M-1}, \tau_{i+M}\}$. For the inner update, we fix $\phi$ and apply multiple steps of gradient descent updates to $\theta$ by minimizing an inner loss $L^{\text{inner}}$. Specifically, for each trajectory $\tau_i \in \mathcal{T}$, we have

$$\Delta\theta_i \propto \nabla_\theta L^{\text{inner}}(\tau_i; \theta_i, \phi) \text{ and } \theta_{i+1} = \theta_i + \Delta\theta_i,$$

where $\nabla_\theta L^{\text{inner}}$ are agent-gradients of $\theta$. By repeating the above process for $M$ times, we get $\theta_i \xrightarrow{\phi} \theta_{i+1} \cdots \xrightarrow{\phi} \theta_{i+M}$. Here, $\theta_{i+M}$ are functions of $\phi$. For simplicity, we abuse the notation and still use $\theta_{i+M}$. Next, we use $\tau_{i+M}$ as a validation trajectory to optimize $\phi$ with an outer loss $L^{\text{outer}}$:

$$\Delta\phi \propto \nabla_\phi L^{\text{outer}}(\tau_{i+M}; \theta_{i+M}, \phi) \text{ and } \phi = \phi + \Delta\phi,$$

where $\nabla_\phi L^{\text{outer}}$ are meta-gradients of $\phi$. Since $\theta_{i+M}$ are functions of $\phi$, we can apply the chain rule to compute meta-gradients $\nabla_\phi L^{\text{outer}}$, with the help of automatic differentiation packages.

## 3 RELATED WORK

Our work is closely related to three areas: optimization in RL, discovering general RL algorithms, and learning to optimize in SL.

### 3.1 OPTIMIZATION IN REINFORCEMENT LEARNING

Henderson et al. (2018) tested different optimizers in RL and pointed out that classical adaptive optimizers may not always consider the complex interactions between RL algorithms and environments. Sarigül & Avci (2018) benchmarked different momentum strategies in deep RL and found that Nesterov momentum is better at generalization. Bengio et al. (2020a) took one step further and showed that unlike SL, momentum in temporal difference (TD) learning becomes doubly stale due to changing parameter updates and bootstrapping. By correcting momentum in TD learning, the sample efficiency is improved in policy evaluation. *These works together indicate that it may not always be appropriate to bring optimization methods in SL directly to RL, without considering the unique properties in RL.* Unlike these works which hand-design new optimizers for RL, we adopt a data-driven approach and apply meta-learning to learn an RL optimizer from data generated in the agent-environment interactions.

### 3.2 DISCOVERING GENERAL REINFORCEMENT LEARNING ALGORITHMS

The data-driven approach is also explored in discovering general RL algorithms. For example, Houthooft et al. (2018) proposed to meta-learn a differentiable loss function which takes the agent's history into account and greatly improve sample efficiency of learning. Similarly, based on DDPG (Lillicrap et al. 2016), Kirsch et al. (2020) proposed MetaGenRL which learns the objective function for deterministic policies using off-policy second-order gradients. Oh et al. (2020) applied meta-learning and discovered an entire update rule for RL by interacting with a set of environments. Instead of discovering the entire update rule, Lu et al. (2022) focused on exploring the mirror learning space with evolution strategies and demonstrated the generalization ability in unseen settings. Bechtle et al. (2021) incorporated additional information at meta-train time into parametric loss functions and extended the idea of learning parametric loss functions from model-free RL to image classification, behavior cloning, and model-based RL. Kirsch et al. (2022) explored the role of symmetries in discovering new RL algorithms and showed that incorporating symmetries can boost generalization ability to unseen learning settings. Jackson et al. (2023) examined the impact of environment design in meta-learning update rules in RL and developed an automatic adversarial environment design approach to improve in-distribution robustness and generalization performance of learned RL algorithms. Following these works, our work adheres to the data-driven spirit, aiming to learning an optimizer instead of general RL algorithms. Moreover, our training and evaluation procedures are largely inspired by them as well.

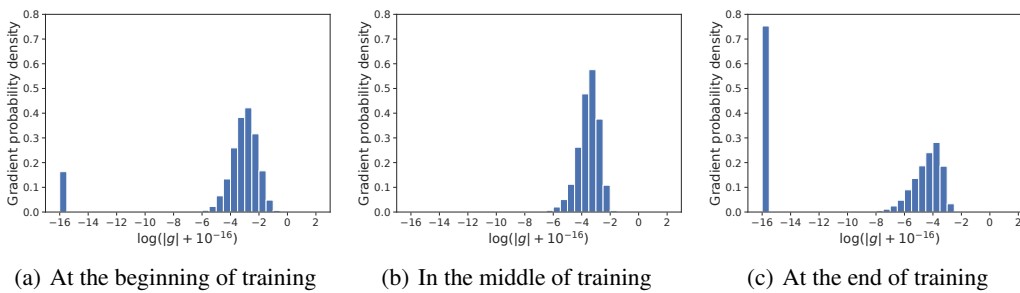

| (a) At the beginning of training | (b) In the middle of training | (c) At the end of training |

Figure 1: Visualizations of agent-gradient distributions (a) at the beginning of training, (b) in the middle of training, and (3) at the end of training. All agent-gradients are collected during training A2C in big_dense_long , optimized by RMSProp. We compute $\log(|g| + 10^{-16})$ to avoid the error of applying $\log$ function to non-positive agent-gradients.

### 3.3   LEARNING TO OPTIMIZE IN SUPERVISED LEARNING

Initially, learning to optimize is only applied to tune the learning rate (Jacobs 1988, Sutton 1992, Mahmood et al. 2012). Recently, researchers started to learn an optimizer completely from scratch. Andrychowicz et al. (2016) implemented learned optimizers with long short-term memory networks (Hochreiter & Schmidhuber 1997) and showed that learned optimizers could generalize to similar and unseen tasks. Li & Malik (2017) applied a guided policy search method to find a good optimizer. Metz et al. (2022a) developed learned optimizers with multi-layer perceptrons which achieve a better balance among memory, computation, and performance.

Learned optimizers are known to be hard to train. Harrison et al. (2022) investigated the training stability of optimization algorithms with tools from dynamical systems and proposed to improve the stability of learned optimizers by adding adaptive nominal terms from Adam (Kingma & Ba 2015) and AggMo (Lucas et al. 2019). Metz et al. (2020a) trained a general-purpose optimizer by training optimizers on thousands of tasks with a large amount of computation. Following the same spirit, Metz et al. (2022b) continued to perform large-scale optimizer training, leveraging more computation ($4,000$ TPU-months) and more diverse SL tasks. The learned optimizer, VeLO, requires no hyperparameter tuning and works well on a wide range of SL tasks. VeLO is the precious outcome of long-time research in the area of learning to optimize, building on the wisdom and effort of many generations. Although marking a milestone for the success of learned optimizers in SL tasks, VeLO still performs poorly in RL tasks, as shown in Section 4.4.4 in Metz et al. (2022b).

The failure of VeLO in RL tasks suggests that designing learned optimizers for RL is still a challenging problem. Unlike previously works that focus on learning optimizers for SL, we aim to learn to optimize for RL. As we will show next, our method is simple, stable, and effective, without using complex neural network structures or incorporating numerous human-designed features. *As far as we know, our work is the first to demonstrate the success of learned optimizers in deep RL.*

## 4   ISSUES IN LEARNING TO OPTIMIZE FOR REINFORCEMENT LEARNING

Learned optimizers for SL are infamously hard to train, suffering from high training instability (Wichrowska et al. 2017, Metz et al. 2019; 2020a, Harrison et al. 2022). In practice, learning an optimizer for RL is even harder (Metz et al. 2022b). In the following, we identify three issues in learning to optimize for RL.

### 4.1   THE AGENT-GRADIENT DISTRIBUTION IS NON-IID

In RL, a learned optimizer takes the agent-gradient $g$ as an input and outputs the agent-parameter update $\Delta\theta$. To investigate the hardness of learning an optimizer for RL, we train an A2C agent in a gridworld (i.e. big_dense_long , see Appendix B for details) with RMSProp (Tieleman & Hinton 2012) and collect the agent-gradients at different training stages. We plot these agent-gradients with logarithmic $x$-axis in Figure 1. The $y$-axis shows the probability mass in each bin. Clearly, the agent-gradient distribution is non-iid, changing throughout the training process. Specifically, at the beginning of training, there are two peaks in the agent-gradient distribution. In the middle of

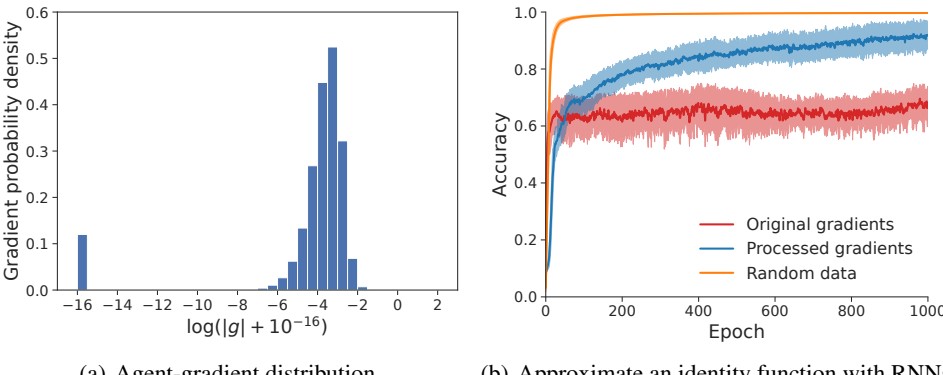

(a) Agent-gradient distribution  (b) Approximate an identity function with RNNs

Figure 2: (a) The agent-gradient distribution of training A2C in big_dense_long . (b) The accuracies of RNN models approximating an identity function with original gradients, processed gradients, and random data as inputs, respectively. Results are averaged over 10 runs, reported with standard errors.

training, most of agent-gradients are non-zero, concentrated around $10^{-3}$. At the end of the training, a large portion of the agent-gradients are zeros. It is well-known that a non-iid input distribution makes training more unstable and reduces learning performance in many settings (Ma et al. 2022, Wang et al. 2023, Khetarpal et al. 2022). Similarly, the violation of the iid assumption would also increase learning instability and decrease efficiency for training learned optimizers. Note that this issue exists in both learning to optimize for supervised learning and RL. However, the agent-gradient distribution from RL is more non-iid than the gradient distribution from supervised learning, since RL tasks are inherently more non-stationary. Please check Appendix E for more details.

## 4.2 RNNs Fail to Approximate an Identity Function with Agent-Gradients as Inputs

In this section, we conduct an experiment to verify if a learned optimizer, represented by a recurrent neural network (RNN) model, can approximate the simplest gradient optimization method — SGD with learning rate 1. In this setting, essentially the RNN model is required to approximate an identity function with agent-gradients as inputs. Surprisingly, the RNN model fails to accomplish this task.

To begin with, similarly to Figure 1, we first train an A2C agent in a gridworld (i.e. big_dense_long ) and collect *all* agent-gradients during training, as shown in Figure 2 (a). We train the RNN model for $1,000$ epochs by minimizing the mean squared error $(\hat{g} - g)^2$, where $g$ the input agent-gradient and $\hat{g}$ is the predicted value. The model makes a correct prediction if $\min((1 - \epsilon')g, (1 + \epsilon')g) \leq \hat{g} \leq \max((1 - \epsilon')g, (1 + \epsilon')g)$ where $\epsilon' = 0.1$; otherwise, $\hat{g}$ is a wrong prediction.

We measure the performance by averaged accuracy over 10 runs, plotted as the red curve in Figure 2 (b). Surprisingly, the accuracy is only about $75\%$, indicating that the RNN fails to approximate the identity function well with agent-gradients as inputs. Note that we observe a similar phenomenon with gradients from supervised learning as inputs (details in Appendix E). To verify that the model has enough expressivity and capacity to represent an identity function, we further train it with randomly generated data where data size is similar to the data size of the agent-gradients. Specifically, the data is sampled uniformly from an interval of fixed length 2, where the lower boundary of this interval is increased during training in order to generate non-iid data. As shown in Figure 2 (b), the training accuracy on this dataset (the orange curve) quickly reaches $100\%$. Together, these results suggest that it is neither non-iid data distribution nor a lack of model expressivity that leads to poor performance.

To conclude, this experiment reconfirms the difficulty of learning good optimizers for RL: given that it is hard for an RNN model to approximate an identity function accurately with agent-gradients as inputs, how could we expect to learn a much more complex parameter update function from scratch?

## 4.3 A Vicious Spiral of Bilevel Optimization

Learning an optimizer while optimizing parameters of a model is a bilevel optimization, suffering from high training instability (Wichrowska et al. 2017, Metz et al. 2020a, Harrison et al. 2022). In

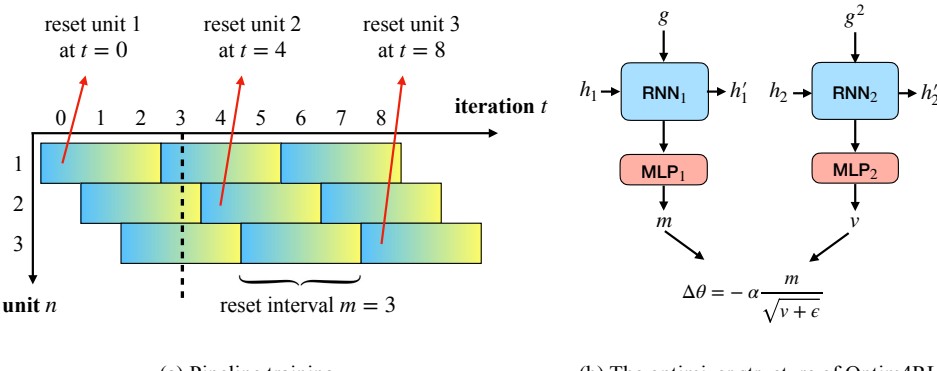

(a) Pipeline training

(b) The optimizer structure of Optim4RL

Figure 3: (a) An example of pipeline training where the reset interval $m = 3$ and the number of units $n = 3$. All training units are reset at regular intervals to diversify training data. (b) The optimizer network structure of Optim4RL. $g$ is the input agent-gradient, $h_i$ and $h_i'$ are hidden states, $\alpha$ is the learning rate, $\epsilon$ is a small positive constant, and $\Delta\theta$ is the parameter update.

RL, due to highly stochastic agent-environment interactions, the agent-gradients have high bias and variance which make the bilevel optimization even more unstable.

Specifically, in SL, it is often assumed that the training set consists of iid samples. However, the input data distribution in RL is non-iid, which makes the whole training process much more unstable and complex, especially when learning to optimize is involved. In most SL settings, true labels are noiseless and stationary (i.e. time-invariant). For example, the true label of a written digit 2 in MNIST (Deng 2012) is $y = 2$ which does not change during training. In RL, TD learning (see Equation (1)) is widely used and TD targets play a similar role as labels in SL. Unlike labels in SL, TD targets are biased, non-stationary, and noisy, due to highly stochastic agent-environment interactions. This leads to a loss landscape that evolves during training and potentially results in the deadly triad (Van Hasselt et al. 2018) and capacity loss (Lyle et al. 2021). Together with biased TD targets, the randomness from state transitions, reward signals, and agent-environment interactions, makes the bias and variance of agent-gradients relatively high. During learning to optimize for RL, meta-gradients are afflicted with large noise induced by the high bias and variance of agent-gradients. With noisy and inaccurate meta-gradients, the improvement of the learned optimizer is unstable and slow. Using a poorly performed optimizer, policy improvement is no longer guaranteed. A poorly performed agent is unlikely to collect "high-quality" data to boost the performance of the agent and the learned optimizer. In the end, this bilevel optimization gets stuck in a vicious spiral: a poor optimizer $\rightarrow$ a poor agent policy $\rightarrow$ collected data of low-quality $\rightarrow$ a poor optimizer $\rightarrow \cdots$.

## 5 OPTIM4RL: A LEARNED OPTIMIZER FOR REINFORCEMENT LEARNING

To overcome the three issues mentioned in Section 4, we propose pipeline training, gradient processing, and a novel optimizer structure. Our optimizer combines all three techniques, named *Optim4RL*, which is more robust and sample-efficient to train in RL than previous methods.

### 5.1 PIPELINE TRAINING

As shown in Figure 1, the gradient distribution of a single agent is non-iid during training. Generally, a good optimizer should be well-functioned under different agent-gradient distributions in the whole training process. To make the agent-gradient distribution more iid, we propose *pipeline training*.

Instead of training only one agent, we train $n$ agents in parallel, each with its own task and optimizer state. Together the three elements form a *training unit* (agent, task, optimizer state); and we have $n$ training units in total. Let $m$ be a positive integer we call the *reset interval*. A complete *training interval* lasts for $m$ training iterations. In Figure 3 (a), we show an example of pipeline training with $m = n = 3$. To train an optimizer effectively, the input of the learned optimizer includes agent-gradients from all $n$ training units. Before training, we choose $n$ integers $\{r_1, \cdots, r_n\}$ such that they are evenly spaced over the interval $[0, m - 1]$. Then we assign $r_i$ to training unit $i$ for $i \in \{1, \cdots, n\}$. At training iteration $t$, we reset training unit $i$ if $r_i \equiv t \pmod{m}$, including

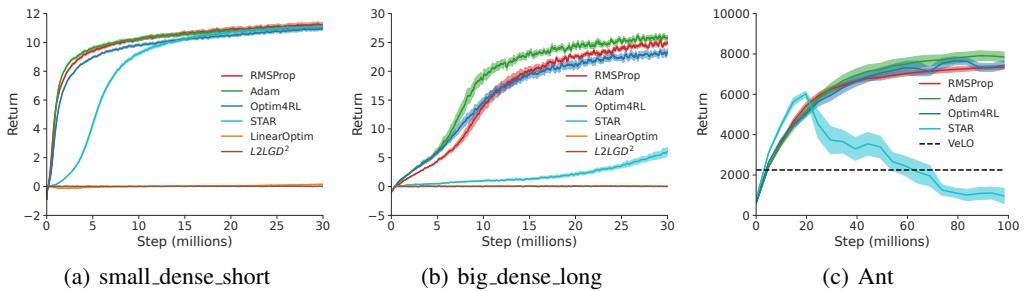

(a) small_dense_short       (b) big_dense_long       (c) Ant

Figure 4: The optimization performance of different optimizers in three RL tasks. Note that the performance of VeLO is estimated based on Figure 11 (a) in Metz et al. (2022b). All other results are averaged over 10 runs and the shaded areas represent standard errors. Optim4RL achieves satisfactory performance in all three tasks which is a significant achievement itself, given that all other learned optimizers fail in big_dense_long.

the agent-parameters, the task, and the optimizer state. By resetting each training unit at regular intervals, it is guaranteed that at any iteration $t$, we can access training data across one training interval. Specifically, in Figure 3 (a), for any training iteration $t \geq 2$, the input consists of agent-gradients from a whole training stage. For instance, at $t = 3$, the input consists of agent-gradients from unit 1 at the beginning of an interval, agent-gradients from unit 2 at the end of an interval, and agent-gradients from unit 3 in the middle of an interval, indicated by the dashed line in Figure 3 (a). With pipeline training, the input agent-gradients are more diverse, spreading across a whole training interval. This makes the input agent-gradient distribution more iid and less time-dependent. Ideally, we expect $m \leq n$ so that the input consists of agent-gradients from all training stages. In our experiments, $n$ is the number of training environments which is task specific; $m$ depends on the training steps of each task and it has a similar magnitude as $n$. We leave the study of the impact of $m$ and $n$ as future work.

## 5.2 GRADIENT PROCESSING

In Section 4.2, we show that an RNN fails to approximate an identity function with agent-gradients as inputs. Figure 2 show that the agent-gradients vary across a wide range in logarithms, from $-\infty$ (we truncate it to $-16$ in the plot) to 0, while their absolute values are in a small range (i.e., $[0, 1]$). This property requires a learned optimizer to distinguish small value differences between various agent-gradients in order to generate accurate parameter updates.

To deal with the particular agent-gradient distribution, before feeding agent-gradients into neural networks, we apply gradient processing to generate better gradient representations. Specifically, for each scalar gradient $g$, we map it to a pair of scalars:

$$g \to [\text{sgn}(g), \log(|g| + \varepsilon)], \tag{2}$$

where $\varepsilon$ is a small value to avoid arithmetic error, e.g. $\varepsilon = 10^{-18}$. By transforming $g$ to $[\text{sgn}(g), \log(|g| + \varepsilon)]$, a neural network can recover $g$ in theory, since this mapping is a one-to-one function. Furthermore, the $\log$ function amplifies the value differences among small agent-gradients, making it easier to distinguish small value differences.

To demonstrate the effectiveness of gradient processing, we apply it and repeat the experiment in Section 4.2 with the same RNN model and training setting. In Figure 2, we plot the training accuracy in the blue curve, averaged over 10 runs. Compared with the result without gradient processing, gradient processing improves the training accuracy from $75\%$ to $90\%$!

## 5.3 IMPROVING THE INDUCTIVE BIAS OF LEARNED OPTIMIZERS

Recently, Harrison et al. (2022) proved that adding adaptive terms to learned optimizers improves the training stability of optimizing a noisy quadratic model. Experimentally, Harrison et al. (2022) showed that adding terms from Adam (Kingma & Ba 2015) and AggMo (Lucas et al. 2019) improves the stability of learned optimizers as well. However, including human-designed features not only makes an optimizer more complex but is also against the spirit of learning to optimize — ideal

learned optimizers should be able to automatically learn useful features, reducing the reliance on human expert knowledge as much as possible. Instead of incorporating terms from adaptive optimizers directly, we design the parameter update function in a similar form to adaptive optimizers:

$$\Delta\theta = -\alpha \frac{m}{\sqrt{v+\epsilon}}, \tag{3}$$

where $\alpha$ is the learning rate, $\epsilon$ is a small positive value to avoid arithmetic error, and $m$ and $v$ are the processed outputs of dual-RNNs, as shown in Figure 3 (b). Specifically, for each input gradient $g$, we generate three scalars $o_1$, $o_2$, and $o_3$. We then set $m = g_{sign} m_{sign} \exp(o_2)$ and $v = \exp(o_3)$, where $g_{sign} \in \{-1, 1\}$ is the sign of $g$ and $m_{sign} \in \{-1, 1\}$ is depended on $o_1$. More details are included in Algorithm 1.

*By parameterizing the parameter update function as Equation* (3)*, we improve the inductive bias of learned optimizers, by choosing a suitable hypothesis space for learned optimizers and reducing the burden of approximating square root and division for neural networks.* In general, we want to learn a good optimizer in a reasonable hypothesis space. It should be large enough to include as many good known optimizers as possible, such as Adam (Kingma & Ba 2015) and RMSProp (Tieleman & Hinton 2012). Meanwhile, it should also rule out bad choices so that a suitable candidate can be found efficiently. An optimizer in the form of Equation (3) meets the two requirements exactly. Moreover, as we show in Section 4.2, it is generally hard for neural networks to approximate mathematical operations (e.g., an identity mapping) accurately (Telgarsky 2017, Yarotsky 2017, Boullé et al. 2020, Lu et al. 2021). With Equation (3), a neural network can spend all its expressivity and capacity learning $m$ and $v$, reducing the burden of approximating square root and division.

Finally, we combine all three techniques and propose our method — a learned optimizer for RL (Optim4RL). Following Andrychowicz et al. (2016), our optimizer also operates coordinatewisely on agent-parameters so that all agent-parameters share the same optimizer. Besides gradients, many previously learned optimizers for SL include human-designed features as inputs, such as moving average of gradient values at multiple timescales, moving average of squared gradients, and Adafactor-style accumulators (Shazeer & Stern 2018). In theory, these features can be learned and stored in the hidden states of RNNs in Optim4RL. So for simplicity, we only consider agent-gradients as inputs. As we will show next, despite its simplicity, our learned optimizer Optim4RL achieves excellent performance in many RL tasks, outperforming several state-of-the-art learned optimizers.

## 6 EXPERIMENT

In this section, we first design experiments to verify that Optim4RL can learn to optimize for RL from scratch. Then we investigate the generalization ability of Optim4RL in different RL tasks. Based on the investigation, we show how to train a general-purpose learned optimizer for RL.

Following Oh et al. (2020), we design six gridworlds: small_dense_long , big_sparse_short , big_sparse_long , big_dense_short , small_dense_short , and big_dense_long . These gridworlds are designed with various properties, different in horizons, reward functions, and state-action spaces. More details are described in Appendix B. Besides gridworlds, we also test our method in several Brax tasks (Freeman et al. 2021). We mainly consider two RL algorithms — A2C (Mnih et al. 2016) and PPO (Schulman et al. 2017). For all experiments, we train A2C in gridworlds and train PPO in Brax tasks. In Optim4RL, we use two GRUs (Cho et al. 2014) with hidden size 8; both multi-layer perceptrons (MLPs) have two hidden layers with size 16. We use Adam to optimize learned optimizers. More implementation details are included in Appendix C.

### 6.1 LEARNING AN OPTIMIZER FOR RL FROM SCRATCH

We first show that it is feasible to train Optim4RL in RL tasks from scratch and that learned optimizers for SL do not work well in RL. We consider both classical and learned optimizers as baselines. Specifically, Adam and RMSProp are two selected classical optimizers. For learned optimizers, we have L2LGD[2] (Andrychowicz et al. 2016), a classical learned optimizer for SL; STAR (Harrison et al. 2022) and VeLO (Metz et al. 2022b), two state-of-the-art learned optimizers for SL.

We select two simple gridworlds (small_dense_short and big_dense_long ) and one complex Brax task (Ant) as test environments. Except for VeLO, we meta-learn optimizers in one task and then test the

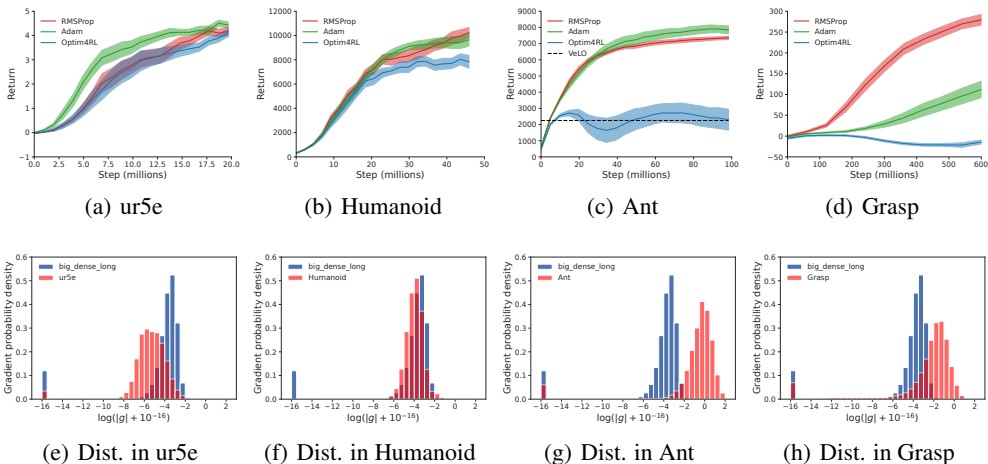

Figure 5: (a-d) The optimization performance of different optimizers in four Brax tasks, averaged over 10 runs. The shaded areas show standard errors. The performance of VeLO is estimated based on Figure 11 (a) in Metz et al. (2022b). (e-h) The agent-gradient distributions in four Brax tasks, compared with agent-gradient distributions in big_dense_long .

fixed learned optimizers in this specific task. The optimization performance of optimizers is measured by returns averaging over 10 runs, as shown in Figure 4. In small_dense_short , Optim4RL and STAR perform pretty well, on par with Adam and RMSProp. However, in big_dense_long , a grid-world with a larger state-action space than small_dense_short , STAR fails to optimize effectively — the return increases slowly as training proceeds. In Ant, STAR's performance is unstable and crashes in the end; Optim4RL performs similarly to Adam and RMSProp while significantly outperforming the state-of-the-art optimizer — VeLO. *While all other learned optimizer fail in big_dense_long , the success of training Optim4RL in RL tasks from scratch is a significant accomplishment in its own right, as it demonstrates the efficacy of our approach and its potential for practical applications.*

**The advantage of the inductive bias of Optim4RL**   To demonstrate the advantage of the inductive bias of OptimRL, we compare OptimRL to an optimizer with another inductive bias as an ablation study. Specifically, we propose a learned optimizer *LinearOptim* which has a "linear" parameter update function: $\Delta\theta = -\alpha(a*g+b)$, where $\alpha$ is the learning rate, $a$ and $b$ are the processed outputs of a RNN model. Note that pipeline training and gradient processing are applied to both optimizers. So the only difference between LinearOptim and Optim4RL is the inductive bias — the parameter update function of LinearOptim is in the form of a linear function while the parameter update function of Optim4RL is inspired by adaptive optimizers (see Equation (3)). As shown in Figure 4, LinearOptim and L2LGD[2] fail to optimize in both small_dense_short and big_dense_long , proving the advantage of the inductive bias of Optim4RL. We also test LinearOptim in Ant but it fails due to NaN errors, probably due to highly non-stationary parameter updates.

**The effectiveness of pipeline training**   Pipeline training is designed to mitigate the problem caused by non-iid agent-gradients. By making the input agent-gradient distribution more iid and less time-dependent, it could improve the training stability and efficiency. To verify this claim, we compare the optimization performance of Optim4RL with and without pipeline training, shown in Table 1. We observe minor performance improvement in two simple tasks (i.e. small_dense_short and big_dense_long ) and more significant performance improvement in two complex tasks (i.e. Ant and Humanoid), confirming the effectiveness of pipeline training.

Table 1: The performance of Optim4RL with and without pipeline training. All results are averaged over 10 runs, reported with standard errors.

| Method \ Task | small_dense_short | big_dense_long | Ant | Humanoid |
|---|---|---|---|---|
| With Pipeline Training | 11.31±0.07 | 23.14±0.33 | 7088±178 | 7658±282 |
| W.o. Pipeline Training | 11.15±0.10 | 23.04±0.41 | 5603±631 | 7053±272 |

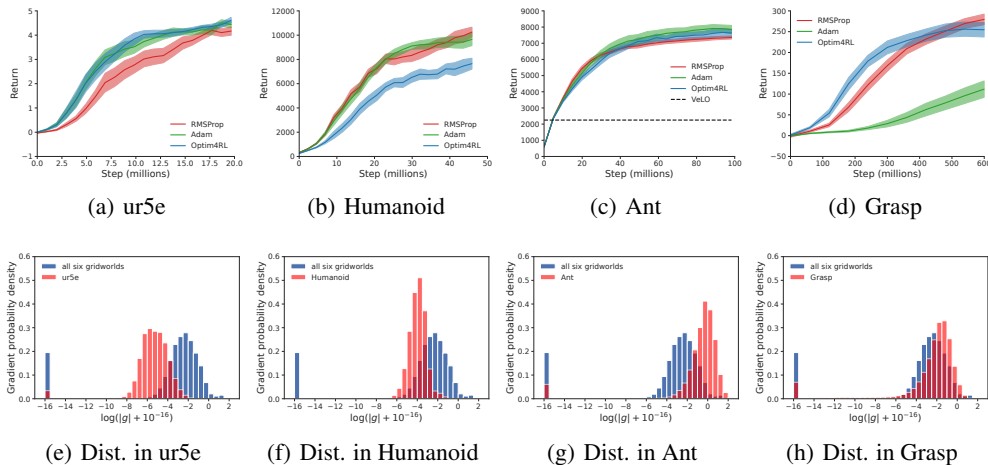

Figure 6: (a-d) Optim4RL shows strong generalization ability and achieves good performance in Brax tasks, although it is only trained in six simple gridworlds from scratch. (e-h) The agent-gradient distributions in four Brax tasks, compared with agent-gradient distributions in all six gridworlds.

## 6.2 TOWARD A GENERAL-PURPOSE LEARNED OPTIMIZER FOR RL

To test the generalization ability of Optim4RL, first we train Optim4RL by running A2C in big_dense_long and then apply Optim4RL to optimize PPO in four Brax tasks. As shown in Figure 5 (a-d), Optim4RL generalizes to ur5e and Humanoid but fails in Ant and Grasp. To investigate, we plot the distribution of all agent-gradients in Figure 5 (e-h). Looking into Figure 5 (e), we observe that there is a large overlap between *the support of agent-gradient distribution* in ur5e and the one in big_dense_long . Same for Humanoid but not for Ant or Grasp. This aligns with the intuition: a learned optimizer is unlikely to perform well given unseen gradients. In other words, *a learned optimizer can only be expected to generalize to tasks with similar agent-gradient distributions*.

The insight points out a promising approach to learning a general-purpose optimizer for RL: meta-train learned optimizers in multiple RL tasks with various agent-gradient distributions. The six gridworlds are designed such that the union of all agent-gradients covers a large range. The learned optimizers are meta-trained in these gridworlds and tested in 8 Brax tasks. We only show the results on 4 tasks in Figure 6 here while other plots are in Figure 7 in Appendix. Compared with Figure 5(e-h), we observe more overlap between the support of agent-gradient distribution of each task and the one for all six gridworlds, especially for Ant and Grasp. Moreover, Optim4RL achieves satisfactory performance in these tasks, showing a better generalization ability compared with Figure 5. Note that Optim4RL surpasses VeLO (the state-of-the-art learned optimizer) in Ant significantly. This is a great success since VeLO is trained for $4,000$ TPU-months on thousands of tasks while Optim4RL is only trained in six toy tasks for a few GPU-hours. *Training a universally applicable learned optimizer for RL tasks is an inherently formidable challenge. Our results demonstrate the generalization ability of Optim4RL in complex unseen tasks which is a great achievement itself, proving the effectiveness of our approach.*

## 7 CONCLUSION AND FUTURE WORK

In this work, we focused on analyzing the hardness of learning to optimize for RL and studied the failures of learned optimizers in RL. Our investigation reveals that agent-gradients in RL are non-iid, vary across a wide range, and have high bias and variance. To mitigate these problems, we introduced pipeline training, gradient processing, and a novel optimizer structure. Combining these techniques, we proposed a learned optimizer for RL, Optim4RL, which can be meta-learned to optimize RL tasks entirely from scratch. Although only trained in toy tasks, Optim4RL showed its strong generalization ability to unseen complex tasks.

Learning to optimize for RL is a challenging problem. Due to memory and computation constraints, our current result is limited since we can only train Optim4RL in a small number of toy tasks. In the

future, by leveraging more computation and memory, we expect to extend our approach to a larger scale and improve the performance of Optim4RL by training in more tasks with diverse RL agents. Moreover, theoretically analyzing the convergence of learned optimizers is also an interesting topic. We hope our analysis and proposed method could inspire and benefit future research, paving the way for better learned optimizers for RL.

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

# A PSEUDOCODE OF OPTIM4RL

The source code will be released upon acceptance. The pseudocode of Optim4RL is presented in Algorithm 1. Specifically, in all our experiments, we use GRUs (Cho et al. 2014) with hidden size 8, MLPs with hidden sizes $[16, 16]$, and ReLU activation functions.

---

**Algorithm 1** Optim4RL: An Optimizer for Reinforcement Learning

---

**Require:** $\text{RNN}_1$ and $\text{RNN}_2$, $\text{MLP}_1$ and $\text{MLP}_2$, hidden states $h_1$ and $h_2$, $\epsilon_1 = 10^{-18}$, $\epsilon_2 = 10^{-18}$, gradient clipping parameter $c = 1$, $b = 1$, a learning rate $\alpha$, input gradient $g$.

  **if** $c > 0$ **then**
    $g \leftarrow \text{clip}(g, -c, c)$                    $\triangleright$ Clip the input gradient
  **end if**
  $g_{sign} \leftarrow \perp [\text{sign}(g)]$             $\triangleright \perp$ denotes the stop-gradient operation
  $g_{log} \leftarrow \perp [\log(|g| + \epsilon_1)]$
  $g_{in} \leftarrow [g_{sign}, g_{log}]$               $\triangleright$ Concatenate $g_{sign}$ and $g_{log}$
  $h_1, x_1 \leftarrow \text{RNN}_1(h_1, g_{in})$ and $o_1, o_2 = \text{MLP}_1(x_1)$
  $m_{sign} = \tanh(o_1 + b)$         $\triangleright$ Add a small bias so that $m_{sign} > 0$ initially
  $m_{sign} = \perp [2(m_{sign} >= 0) - 1 - m_{sign}] + m_{sign}$   $\triangleright$ Use straight-through s.t. $m_{sign} \in \{-1, 1\}$
  $m = g_{sign} m_{sign} \exp(o_2)$        $\triangleright$ Compute $m$: 1st pseudo moment estimate
  $h_2, x_2 \leftarrow \text{RNN}_2(h_2, 2g_{log})$ and $o_3 = \text{MLP}_2(x_2)$
  $v = \exp(o_3)$                $\triangleright$ Compute $v$: 2nd pseudo moment estimate
  $\Delta\theta \leftarrow -\alpha \frac{m}{\sqrt{v} + \epsilon_2}$           $\triangleright$ Compute the parameter update

---

# B GRIDWORLDS

We follow Oh et al. (2020) and design 6 gridwolds. In each gridworld, there are $N$ objects. Each object is described as $[r, \epsilon_{\text{term}}, \epsilon_{\text{respawn}}]$. Object locations are randomly determined at the beginning of each episode, and an object reappears at a random location after being collected, with a probability of $\epsilon_{\text{respawn}}$ for each time-step. The observation consists of a tensor $\{0, 1\}^{N \times H \times W}$, where $N$ is the number of objects, and $H \times W$ is the size of the grid. An agent has 9 movement actions for adjacent positions, including staying in the same position. When the agent collects an object, it receives the corresponding reward $r$, and the episode terminates with a probability of $\epsilon_{\text{term}}$ associated with the object. In Table 2 – Table 7, we describe each gridworld in detail.

Table 2: small_dense_short

| Component | Description |
|---|---|
| Size ($H \times W$) | $4 \times 6$ |
| Objects | $[100.0, 0.0, 0.5], [-100.0, 0.5, 0.5]$ |
| Horizon | 50 |

Table 3: small_dense_long

| Component | Description |
|---|---|
| Size ($H \times W$) | $6 \times 4$ |
| Objects | $[1000.0, 0.0, 0.5], [-1000.0, 0.5, 0.5]$ |
| Horizon | 500 |

# C EXPERIMENTAL DETAILS

In this work, we apply Jax (Bradbury et al. 2018) to do automatic differentiation. Unless mentioned explicitly, we use ReLU as the activation function. For A2C training in gridworlds, the feature net

Table 4: big_sparse_short

| Component | Description |
| --- | --- |
| Size ($H \times W$) | $10 \times 12$ |
| Objects | $2 \times [100.0, 0.0, 0.05], 2 \times [-100.0, 0.5, 0.05]$ |
| Horizon | 50 |

Table 5: big_sparse_long

| Component | Description |
| --- | --- |
| Size ($H \times W$) | $12 \times 10$ |
| Objects | $2 \times [10.0, 0.0, 0.05], 2 \times [-10.0, 0.5, 0.05]$ |
| Horizon | 500 |

is an MLP with hidden size 32 for the "small" gridworlds. For the "big" gridworlds, the feature net is a convolution neural network (CNN) with 16 features and kernel size 2, followed by an MLP with output size 32. We set $\lambda = 0.95$ to compute $\lambda$-returns. The discount factor $\gamma = 0.995$. One rollout has 20 steps. The critic loss weight is $0.5$ and the entropy weight is $0.01$. For PPO training in Brax games, we use the same settings in Brax examples [2]. For A2C, the inner loss is the standard A2C loss, while the outer loss is the actor loss in A2C loss. For PPO, both the inner loss and outer loss are the standard PPO loss. To meta-learn optimizers, we set $M = 4$ in all experiments; that is, for every outer update, we do 4 inner updates. Potentially, larger $M$ could lead to more farsighted learning but results in increasing memory and computation requirement. We set $M = 4$ as a trade-off which also works well in practice. Following common practice (Lu et al. 2022), we report results averaged over 10 runs.

Other details are presented in the following sections.

## C.1 COMPUTATION RESOURCES

All our experiments can be trained with V100 GPUs. For some experiments, we use 4 V100 GPUs due to a large GPU memory requirement. The computation to repeat all experimental results in this work is not high (less 10 GPU years). However, the exact used computation is hard to estimate.

## C.2 IMPLEMENTATION DETAILS FOR SECTION 4

We collect agent-gradients by training A2C in big_dense_long for $30M$ steps with learning rate $3e - 3$, optimized by RMSProp. All collected agent-gradients are divided into 30 parts by time-steps. We then plot the agent-gradients in the first, sixteenth, last part as the agent-gradient distributions at the beginning of training, in the middle of training, and at the end of training respectively. To approximate the identity function, the RNN model is an LSTM (Hochreiter & Schmidhuber 1997) with hidden size 8, followed by an MLP with hidden sizes $[16, 16]$. We train the RNN model for $1,000$ epochs and pick the best learning rate from $\{3e - 3, 1e - 3, 3e - 4, 1e - 4\}$.

## C.3 IMPLEMENTATION DETAILS FOR SECTION 6.1

For both LinearOptim and L2LGD[2], the model consists a GRU with hidden size 8, followed by an MLP with hidden sizes $[16, 16]$. For STAR, we use the official implementation from learned_optimization [3].

The agent learning rate in small_dense_short and big_dense_long are set to $1e - 2$ and $3e - 3$, respectively. The number of environments / training units $n$ is 512. The reset interval $m$ is chosen from $[256, 512]$. We use Adam as the meta optimizer and choose the meta learning rate from

---

[2]https://github.com/google/brax/blob/main/notebooks/training.ipynb
[3]https://github.com/google/learned_optimization/blob/main/learned_optimization/learned_optimizers/adafac_nominal.py

Table 6: big_dense_short

| Component | Description |
|---|---|
| Size ($H \times W$) | $9 \times 13$ |
| Objects | $2 \times [10.0, 0.0, 0.5]$, $2 \times [-10.0, 0.5, 0.5]$ |
| Horizon | 50 |

Table 7: big_dense_long

| Component | Description |
|---|---|
| Size ($H \times W$) | $13 \times 9$ |
| Objects | $2 \times [1.0, 0.0, 0.5]$, $2 \times [-1.0, 0.5, 0.5]$ |
| Horizon | 500 |

$\{3e-5, 1e-4, 3e-4, 1e-3\}$. For STAR, we pick step multiplicator from $\{3e-3, 1e-3, 3e-4\}$, nominal step-size from $\{3e-3, 1e-3, 3e-4, 0.0\}$, and weight decay from $\{0.0, 0.1, 0.5\}$. For Adam and RMSProp, we pick the best agent learning rate from $\{3e-2, 1e-2, 3e-3, 1e-3, 3e-4, 1e-4\}$.

For training Optim4RL in Ant from scratch, in order to reduce memory requirement, we set the number of mini-batches from the default value 32 to 8. The hidden sizes of the value network are changed from $[256, 256, 256, 256, 256]$ to $[64, 64, 64, 64, 64]$. The number of environments / training units $n$ is 2048. The reset interval $m$ is chosen from $\{512, 1024\}$.

## C.4 IMPLEMENTATION DETAILS FOR SECTION 6.2

Optim4RL is meta-trained in 6 gridworlds and then tested in 8 gridworlds. We use Adam as the meta optimizer and choose the meta learning rate from $\{3e-5, 1e-4, 3e-4, 1e-3\}$. The number of environments / training units $n$ is 512. The reset interval $m$ is chosen from $\{256, 512\}$. In Figure 7, we present results in 8 Brax games.

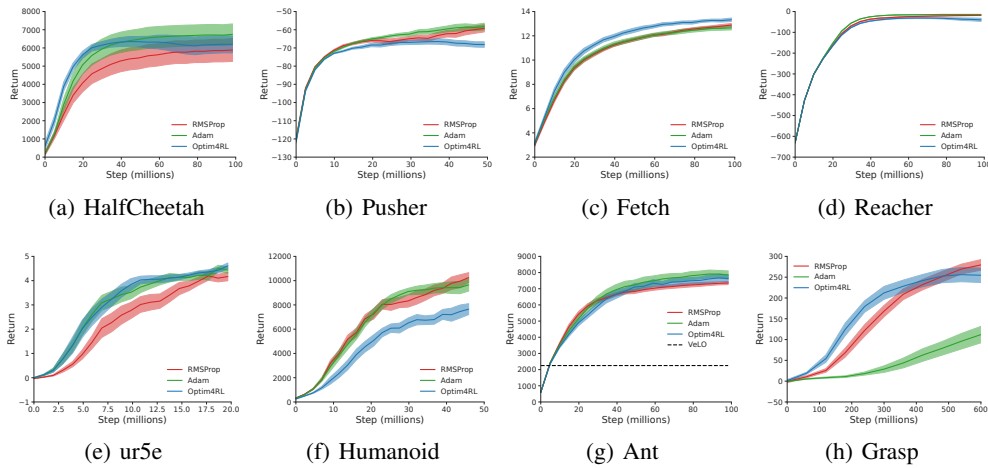

Figure 7: Optim4RL shows strong generalization ability and achieves good performance in Brax tasks, although it is only trained in six simple gridworlds from scratch. For comparison, VeLO (Metz et al. 2022b) is trained for $4,000$ TPU-months with thousands of tasks but only achieves sub-optimal performance in Ant. The above results demonstrate the generalization ability of Optim4RL in complex unseen tasks which is a significant achievement itself, proving the effectiveness of our approach.

# D    ROBUSTNESS TO DIFFERENT HYPER-PARAMETER SETTINGS

In this section, we show that Optim4RL not only generalizes to unseen tasks, but also transfers to different hyper-parameter settings. To be specific, we test our learned optimizer Optim4RL under different hyper-parameter settings in two gridworlds – small_dense_short and big_dense_long . We report the returns at the end of training, averaged over 10 runs. As shown in Table 8, Table 9, and Table 10, Optim4RL is robust under different hyper-parameter settings, such as GAE $\lambda$, entropy weight, and discount factor in PPO.

Table 8: The performance of Optim4RL with different *GAE $\lambda$* values in two gridworlds. All results are averaged over 10 runs, reported with standard errors.

| Task | Parameter Value | Return |
|------|-----------------|--------|
| small_dense_short | 0.9 | 11.51±0.11 |
| small_dense_short | 0.95 | 11.25±0.09 |
| small_dense_short | 0.99 | 10.81±0.10 |
| small_dense_short | 0.995 | 10.66±0.10 |
| big_dense_long | 0.9 | 23.35±0.44 |
| big_dense_long | 0.95 | 23.57±0.35 |
| big_dense_long | 0.99 | 21.31±0.37 |
| big_dense_long | 0.995 | 20.55±0.31 |

Table 9: The performance of Optim4RL with different *entropy weights* in two gridworlds. All results are averaged over 10 runs, reported with standard errors.

| Task | Parameter Value | Return |
|------|-----------------|--------|
| small_dense_short | 0.005 | 11.01±0.09 |
| small_dense_short | 0.01 | 11.13±0.05 |
| small_dense_short | 0.02 | 11.29±0.07 |
| small_dense_short | 0.04 | 11.25±0.09 |
| big_dense_long | 0.005 | 22.41±0.34 |
| big_dense_long | 0.01 | 22.45±0.46 |
| big_dense_long | 0.02 | 22.59±0.25 |
| big_dense_long | 0.04 | 19.96±0.75 |

Table 10: The performance of Optim4RL with different *discount factors* in two gridworlds. All results are averaged over 10 runs, reported with standard errors.

| Task | Parameter Value | Return |
|------|-----------------|--------|
| small_dense_short | 0.9 | 12.47±0.08 |
| small_dense_short | 0.95 | 12.32±0.05 |
| small_dense_short | 0.99 | 11.48±0.05 |
| small_dense_short | 0.995 | 11.01±0.10 |
| big_dense_long | 0.9 | 18.13±1.89 |
| big_dense_long | 0.95 | 25.01±0.82 |
| big_dense_long | 0.99 | 25.45±0.27 |
| big_dense_long | 0.995 | 22.07±0.47 |

# E    THE EXPERIMENTS FOR THE GRADIENT DISTRIBUTION IN SUPERVISED LEARNING

In this section, we show that the gradient distribution in supervised learning is also non-iid but it is more iid than the agent-gradient distribution in RL. Specifically, we train a neural network on

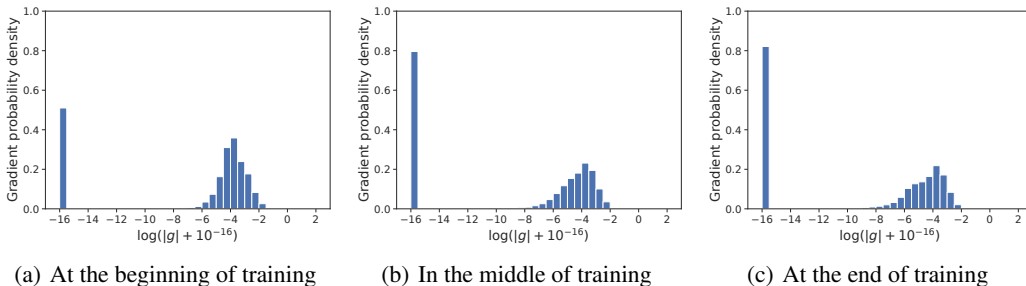

(a) At the beginning of training    (b) In the middle of training    (c) At the end of training

Figure 8: Visualizations of gradient distributions (a) at the beginning of training, (b) in the middle of training, and (3) at the end of training. All gradients are collected during training in MNIST, optimized by RMSProp. We compute $\log(|g| + 10^{-16})$ to avoid the error of applying $\log$ function to non-positive agent-gradients.

MNIST (Deng 2012) for 10 epochs with RMSProp and collect gradients at different training stages. Note that the network is the same as the actor network used in training A2C in big_dense_long , except for the output layer. We plot these gradients with logarithmic $x$-axis in Figure 8. Similar to Figure 1, the gradient distribution is also non-iid, changing throughout the training process.

To show the gradient distribution from training on MNIST is more iid than the agent-gradient distribution from training in big_dense_long , we compute the Wasserstein distance (WD) between the (agent-)gradient distribution at different training stages and the distribution of all (agent-)gradients during training in Table 11. Note that a smaller distance indicates a higher iid degree. Thus these results support the above claim.

Table 11: The Wasserstein distance between the (agent-)gradient distribution at different training stages and the distribution of all (agent-)gradients during training.

| Task                          WD Value | WD(beginning, all) | WD(middle, all) | WD(end, all) |
|---|---|---|---|
| MNIST | $4.0642 \times 10^{-4}$ | $0.7205 \times 10^{-4}$ | $1.4204 \times 10^{-4}$ |
| big_dense_long | $7.1583 \times 10^{-4}$ | $1.1726 \times 10^{-4}$ | $6.7967 \times 10^{-4}$ |

In Figure 9, we visualize the distribution of all gradients during training on MNIST and plot the accuracy curves of approximating an identity function with gradients in MNIST as inputs. The results are similar to Figure 2. Specifically, without gradient processing, the accuracy is only about 30%; with gradient processing, the accuracy is much higher, around 80%.

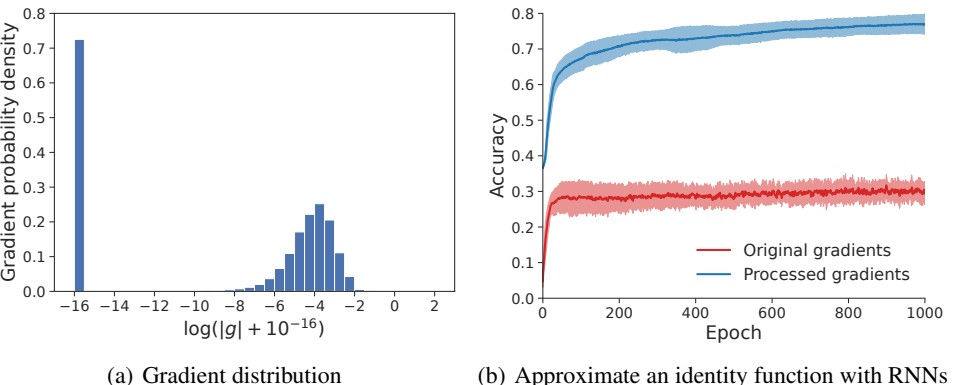

(a) Gradient distribution

(b) Approximate an identity function with RNNs

Figure 9: (a) The gradient distribution of training on MNIST. (b) The accuracies of RNN models approximating an identity function with original gradients and processed gradients as inputs, respectively. Results are averaged over 10 runs, reported with standard errors.

