# OpenReview forum: "Learning to Optimize for Reinforcement Learning"
_ICLR.cc/2024/Conference — Submitted to ICLR 2024_

### Official Review · Reviewer_wz35 · 2023-10-29

**Soundness:** 2 fair
**Presentation:** 4 excellent
**Contribution:** 2 fair
**Rating:** 3
**Confidence:** 5

**Summary:**

The paper presents a method for meta-learned optimization in RL, named Optim4RL. This consists of three components - pipeline training, gradient transformations, and an update formulation - which are designed to tackle specific issues in this setting. The evaluation investigates the performance of this method when meta-trained on simple grid-world environments and evaluated on much more challenging environments, primarily evaluating against Adam and RMSProp, in addition to a collection of alternative meta-learned optimizers on in-distribution tasks.

I am recommending rejection for this paper, primarily due to the lack of ablations. However, I would be very willing to increase my score if the suggested ablations were performed, in addition to a discussion of the wider literature for this problem setting.

**Strengths:**

1. Related work extensively covers RL optimization and meta-optimization literature.
2. The paper is impressively well-written and structured. Section 4 is particularly well-structured, presenting a clear set of hypotheses about the problems with meta-optimization in RL.
3. Each proposed component is simple, but clearly motivated and presented in Section 5 and Figure 3.
4. Many related methods are compared against Optim4RL in Figure 4, however, these could be evaluated further (see weaknesses).

**Weaknesses:**

1. The predominant flaw with this paper is the evaluation of the proposed components. Section 5 is highly systematic in motivating each problem, before proposing a component as a solution. However, the evaluation does not ablate the components, making it impossible to discern their individual impact. The exception to this is LinearOptim, which is an ablation of the proposed inductive bias, however, I believe this should be highlighted. Whilst the comparison to existing baselines is interesting, this is a __fundamental__ requirement when evaluating a model composed of multiple novel components.
2. The presentation of the results could be clearer for drawing conclusions. Whilst training curves are useful, they make it difficult to quantitatively determine significance.
3. The wide range of baselines in Figure 4 is good, but it is unclear why these are not carried forward for the remainder of the evaluation. Given that meta-training is the largest computational cost, these shouldn't be out-of-budget to run. In particular, STAR is performative enough that it is plausible it would achieve competitive performance on the remaining tasks, which should be investigated.
4. The inability of LinearOptim to learn anything is very surprising and should be investigated further to ensure it is not erroneous.
5. There is a broad base of related work on meta-learned RL objective functions, which is not discussed or compared against. While these are a different class of inductive bias, they are solving the same problem as the class of meta-optimizers discussed here. Notably, the evaluation procedure and environments are from Oh et al. (2020), a learned objective algorithm, but it is not compared against. While it is understandable to not compare Optim4RL against all of these methods, they should at least be discussed as alternative approaches to the same problem in the related work. Namely, EPG (Houthooft et al., 2018), LPG (Oh et al., 2020), MetaGenRL (Kirsch et al., 2020), ML^3 (Bechtle et al., 2021), SymLA (Kirsch et al., 2022), DPO (Lu et al., 2022), GROOVE (Jackson et al., 2023).
6. Many of the claims are misleading or ambiguous. In conclusion, the claim that Optim4RL is "the first learned optimizer that can be meta-learned to optimize RL tasks entirely from scratch" is confusing, since all existing learned optimizers can be and are applied to RL in this paper. If this is intended to claim this is the first meta-learned optimizer designed for RL, then the omission meta-learned objective function literature becomes even more apparent, since there is extensive work solving the same problem for RL.

**Questions:**

1. Major typo in Algorithm 1: the sign and magnitude of your update are both computed from o_1. I assume this is a typo since, if correct, your method would only be capable of outputting large positive updates and small negative updates.
2. Transformation of the gradient output is a major component of gradient processing, but only the input transformations are discussed in the main body. An expanded form of the output could be presented in the main body.
3. In Figure 5, you suggest the overlap in the support of the gradients is a predictor of performance. A simple experiment to evaluate this would be retraining the optimizer with rescaled rewards on the grid-world tasks, which would shift the support of the meta-training gradients. If this improved performance, this would significantly strengthen the hypothesis.

---

> ### Author Response · Authors · 2023-11-17
> **Reply to Reviewer wz35**
>
> Thank you for your detailed feedback, suggestions, and comments. We address your points in the following.
> - **Add more ablation studies.** Please check the general response.
> - **In Figure 5, you suggest the overlap in the support of the gradients is a predictor of performance. A simple experiment to evaluate this would be retraining the optimizer with rescaled rewards on the grid-world tasks, which would shift the support of the meta-training gradients.** Yes, this is exactly what we did. We designed six gridworlds with different reward scales such that the union of all agent-gradients covers a large range. With carefully rescaled rewards, we can significantly improve the performance of the learned optimizer trained in gridworlds for Brax tasks. In fact, simply rescaling the agent-gradients before feeding them into the optimizer also works.
> - **The inability of LinearOptim to learn anything is very surprising and should be investigated further to ensure it is not erroneous.** We checked the implementation again and ensured it was not erroneous.
> - **Transformation of the gradient output is a major component of gradient processing. An expanded form of the output should be presented.** We will add more explanations for this part.
> - **Add more citations, modify misleading claims, and correct typos.** We will add those citations and fix the typos and misleading claims.

---

> ### Comment · Reviewer_wz35 · 2023-11-19
>
> Thank you for your response and new experiments.
>
> **New ablation studies** - Thank you for including these ablations. Regarding *pipeline training*, the results are somewhat promising but I would like to see them strengthed with the full environment set and labeled significance. Regarding *optimizer structure*, it is not clear from the experimental setup that Figure 4 shows an ablation of this specifically, since Optim4RL is proposed as a combination of methods. This should be clearly described in the experimental setup or Figure 4. Regarding *gradient preprocessing*, it is quite significant for the story of this paper that this method does not aid (and possibly harms) performance. I commend the authors for being candid about these results and the paper still contains useful contributions without them, however incorporating this and the other new results would require a *significant revision*, which is likely out of scope for this rebuttal period.
>
> **Gradient distribution experiments** - Thank you for pointing this out. Even though I agree with the motivation for this experiment, it is not clearly labeled in the paper (see fig 5, 6 captions and section 6.2 title) and incomplete without replotting the agent-gradient distribution in figure 6. Furthermore, adding training environments (rather than just rescaling the reward magnitude on the original environment) adds a confounder beyond the gradient distribution. I would strongly recommend merging figures 5 and 6, such that figure 5 a-d contains one more curve for the Optim4RL on extended training set and 5 e-h contains the distribution over the extended training set.
>
> **LinearOptim results** - I was suggesting that the results be extended with further environments, since they are highly surprising and necessitate further investigation. The release of anonymized source code would also be reassuring.
>
> **Remaining edits** - Thank you for being receptive to these suggestions - I would like to see them included in the revision.

---

> > ### Author Response · Authors · 2023-11-20
> > **Further reply**
> >
> > Thank you for your reply.
> > - **The ablation of pipeline training**:  The current ablation results already support the effectiveness of this method. Due to high computation requirements, it is very unlikely to finish the experiments of the full environment set in time.
> > - **Gradient distribution experiments**: We will replot the agent-gradient distribution in Figure 6, which is simply a union of all agent-gradients in 6 gridworlds.
> > - **LinearOptim results**: We are meta-training LinearOptim in Ant and will report results if we can get them in time.

---

> > > ### Comment · Reviewer_wz35 · 2023-11-21
> > >
> > > Thank you for your response.
> > >
> > > I am currently maintaining my score, but I am inclined to raise it after reviewing the new revision. I believe the paper will be a strong submission once the promised edits and new results are implemented, but I do not believe it is in an acceptable state prior to this.
> > >
> > > This is the paper I was referring to: Jackson, Matthew Thomas, et al. "Discovering General Reinforcement Learning Algorithms with Adversarial Environment Design." Thirty-seventh Conference on Neural Information Processing Systems. 2023.

---

> ### Author Response · Authors · 2023-11-21
> **Question about related work**
>
> Dear Reviewer,
>
> We did not find the mentioned related work GROOVE (Jackson et al., 2023). Could you please provide more details about this work?
>
> Thanks.

---

### Official Review · Reviewer_8FzQ · 2023-10-31

**Soundness:** 3 good
**Presentation:** 3 good
**Contribution:** 3 good
**Rating:** 6
**Confidence:** 4

**Summary:**

The authors propose a meta-learning procedure for learning optimizers for reinforcement learning called Optim4RL. Their method has the following key components:
- Pipeline training: Use multiple agents each at different training stages (early/mid/late) to make data distribution (more) stationary.
- Gradient pre-processing: Transform the gradient so that the input is sensitive to changes of the gradient in (approximately) log space.
- Inductive bias: Structure the update using a form similar to Adam, providing good inductive bias

In their experiments Optim4RL is shown to (1) outperform existing learnt optimizers, and (2) generalise to problems outside of the training set.

**Strengths:**

- Each of the parts of the proposed learnt optimizer solve important problems in meta-learning optimizers, and are well motivated.
- The toy problem where the optimizer has to learn the identity function is simple and informative.
- The paper has informative analysis on the gradient distribution (e.g. I like the plots in Figure 5 visualising the train-test marginal distribution of gradients).
- Achieves generalisation to different tasks (Brax) from simple grid based problems.
- Generally the paper is clear and easy to follow.

**Weaknesses:**

- The optimizer achieves marginally worse performance than Adam on the tasks that it is meta-trained on. It seems like the learnt optimizer should at least be able to "overfit" to the training task to outperform Adam here.
- On unseen tasks the optimizer is significantly worse than Adam.
- The training and test tasks are relatively toy problems (the authors do acknowledge this weaknesses).

**Questions:**

- The below text confused me - why do we need to check that the model has enough capacity to represent the identity function (e.g. a single linear layer can represent the identity function easily)?
Also, why do we need an RNN on this problem (is the input not just the current gradient?)?

> To verify that the model has enough expressiveness and capacity to represent an identity function, we further train it with randomly generated data where data size is similar to the data size of the agent-gradients.

- Transforming the gradient passed to the RNN into a richer representation makes sense. Additionally this seems to help a lot in terms of performance so it seems worth digging deeper into. Were other transformations tried - e.g. fourier feature embedding?

- Would it be possible to add the STAR benchmark to Figure 4 (ant), and for Table 5? This would allow us to see how well Optim4RL generalizes relative to another learnt optimizer.

- In figure 4 (b) it seems like STAR is starting to learn a bit. Is it possible that with a bit more hyper-parameter tuning it would match the other optimizers in performance?

I acknowledge that a lot of my questions require more compute, and that this is a very compute heavy task. I do think that these would significantly strengthen the paper - as they would help make the results more decisive.

---

> ### Author Response · Authors · 2023-11-17
> **Reply to Reviewer 8FzQ**
>
> Thank you for your detailed feedback, suggestions, and comments. We address your points in the following.
> - **Add more ablation studies.** Please check the general response.
> - **Why can’t the learned optimizer "overfit" to the training task to outperform Adam?** Note that we choose a very small network to represent the learned optimizer so that we can train it within the computation and memory budget. This also helps reduce the deployment computation burden, i.e. reducing the inference computation of a fixed learned optimizer. However, the disadvantage of using a small network is that it is hard for the learned optimizer to “overfit“ a relatively complex task, such as the gridworlds which have image inputs. Instead, we did experiments on a relatively simple task — Catch form [bsuite](​​https://github.com/google-deepmind/bsuite). The results indeed showed that the learned optimizer can “overfit” Catch, converging faster than Adam and RMSProp.
> - **Why do we need to check that the model has enough capacity to represent the identity function (e.g. a single linear layer can represent the identity function easily)? Why do we need an RNN for this problem? Isn’t the input just the current gradient?** We want to verify whether a learned optimizer represented by an RNN model can approximate the simplest gradient optimization method — SGD with learning rate 1. In this setting, essentially the RNN model is required to approximate an identity function with agent-gradients as inputs. We use RNN models so that historical gradients can be taken into account and are not completely ignored. For example, historical gradients are important to compute the momentum in RMSProp. Theoretically, a neural network can represent an identity function. However, in practice, it might not be true. Please check [this paper](http://arxiv.org/abs/1902.04698) for an example.
> - **In Figure 4 (b) it seems like STAR is starting to learn a bit. Is it possible that with a bit more hyper-parameter tuning it would match the other optimizers in performance?** We tuned STAR with more hyper-parameter options and improved its performance (average return) to 10; however, this is still much worse than the performance of our method (average return>20).

---

> > ### Comment · Reviewer_8FzQ · 2023-11-20
> >
> > Thanks for the clarification and updated results. I will stick to my original recommendation.

---

### Official Review · Reviewer_tcjw · 2023-10-31

**Soundness:** 3 good
**Presentation:** 3 good
**Contribution:** 3 good
**Rating:** 6
**Confidence:** 3

**Summary:**

The authors propose a new learned optimizer, Optim4RL, to address the challenges of using learned optimizers in RL.

While learned optimization has shown benefits in the supervised learning community, SOTA optimizers for Supervised Learning (SL) fail in the RL setting. The authors investigate this phenomenon by analyzing the distribution of the gradients of agent parameters at the start, middle, and end of training. Through this analysis, the authors demonstrate that the gradients are non-I.I.D. Moreover, the absolute values of the gradients lie in a small range. The authors then demonstrate the difficulty of the RNN module -- commonly used in the learned optimizer -- to approximate an identity function using the gradient data (75% accuracy). Using this analysis, they underscore the bias and variance in the gradients as a key issue that makes learned optimization hard in RL and argue that this is further exacerbated by the bi-level optimization in learned optimizers. (poor optimizer -- poor policy -- lower quality data)

The authors then propose three key ways to mitigate these issues:
- Gradient Processing: a 1-1- mapping that uses a log transformation to magnify absolute value differences between small gradients to mitigate the logarithmic gradient variation. This boosts the accuracy of the RNN on the identity task from 75% to 90%
- Pipeline Training: add diversity to the gradient inputs to the learned optimizer through a distributed training regime by parallelly training multiple agents being reset at different periods and using all of their gradients for the learned optimizer. This mitigates the non-iid nature of the data since data now comes from different points of training
- Biasing the optimizer: Building on the analysis of [Harrison et al., 2022], they utilize both the gradient and its squared value as inputs to two RNNs. This mitigates the need to approximate square roots and division by the learned optimizer and stabilizes the meta-update

The combination of these three components -- Optim4RL -- demonstrates improved stability and effectiveness in optimizing RL tasks compared to baselines of hand-designed optimizers (Adam and RMSProp), learned optimizers for Supervised learning (L2LGD$^2$, STAR, and VeLO), and linear parameter update instead of a squared

[Harrison et al., 2022] Harrison, J., Metz, L., & Sohl-Dickstein, J. (2022). A closer look at learned optimization: Stability, robustness, and inductive biases. Advances in Neural Information Processing Systems, 35, 3758-3773.

**Strengths:**

### Originality
The paper tackles a novel direction of RL-specific learned optimization by looking deeper into what kind of RL-specific inputs need to be adapted.

### Quality
The work is insightful and generally conducted comprehensively.

### Clarity
The paper is written clearly and understandably. Overall, the presentation is clear and well done.


### Significance
The research direction is significant since learned optimization is yet to take hold in RL properly and is very important if achieved.

**Weaknesses:**

There seem to be a lot of central design decisions/hyperparameters in the training procedure that are not justified:
- 4 inner updated per outer update
- The decision to average returns over ten runs
- The threshold for gradient processing
- Epsilon in the parameter update

The agglomerative procedure to incorporate diversity in the gradient distribution seems not fully ablated. See my questions on this for further details.

I am unsure if 10 GPU years is a realistically feasible budget for most practitioners. One of the issues with learned optimization in SL has been this exact problem. I think commentary on how to bring this cost down would be highly beneficial for hte community, especially given the recent surge in JAX-based parallelization with developments such as PureJAXRL (https://github.com/luchris429/purejaxrl).

**Questions:**

- What happens when we don't do individual pre-processing steps?  -- Are there any ablations that demonstrate the effectiveness of individual modifications?
- What constitutes the middle of training? is it the same for each environment or different across environments?
- How many seeds were the experiments reported on? How did the authors determine them?
- Resetting provides the optimizer data at different training stages. Have the authors analyzed how different values of m and n impact this? Do we require them to be equal all the time?
- Given that pipeline training can be computationally expensive, have the authors examined methods to extract maximum benefit from this procedure? For example, could reset times be adapted by leveraging optimizer reset properties? [Asadi et al., 2023]
- Does the learned optimizer mitigate the requirement for dynamic hyperparameter optimization [Mohan et al., 2023]? To what extent is the problem addressed in this work related to the AutoRL problem, given that there are still optimizer-related hyperparameters?

[Asadi et al., 2023]  Asadi, K., Fakoor, R., & Sabach, S. (2023). Resetting the Optimizer in Deep RL: An Empirical Study. arXiv preprint arXiv:2306.17833.

[Mohan et al, 2023] Mohan, A., Benjamins, C., Wienecke, K., Dockhorn, A., & Lindauer, M. (2023). AutoRL Hyperparameter Landscapes. AutoML Conference 2023 (https://openreview.net/forum?id=Ec09TcV_HKq).

---

> ### Author Response · Authors · 2023-11-17
> **Reply to Reviewer tcjw**
>
> Thank you for your detailed feedback, suggestions, and comments. We address your points in the following.
> - **Add more ablation studies.** Please check the general response.
> - **Add more citations and training details.** We will add those citations and more training details to clarify.
> - **Have the authors analyzed how different values of m and n impact this? Could reset times be adapted by leveraging optimizer reset properties? Do we require them to be equal all the time?** m and n are not required to be equal all the time. In our experiments, n is the number of training environments that depends on the task itself; m is affected by the training steps of each task and it has a similar magnitude as n. We leave the study of the performance impact of m and n as future work.

---

> > ### Comment · Reviewer_tcjw · 2023-11-20
> > **Reply to authors**
> >
> > Thank you for the response and the clarifications!
> >
> > - **Ablation studies:** The ablations are extremely helpful in demonstrating the impact of gradient preprocessing and the inductive bias.
> > - **Impact of m and n:** Thank you for clarifying that point. Learning optimizers can help ease or alleviate the need for highly task-specific hyperparameter optimization. However, in the experimental setup, the authors introduce two new task-specific hyperparameters that can impact the diversity of data in the shared buffer and the training speed of the optimizer. Therefore, whether these require precise tuning for each task or whether general values such as m=3 and n=3 can work across tasks is relevant for anyone hoping to use this optimizer. A discussion about this point would be highly beneficial in strengthening the usability of this optimizer.
> > - **Related Work:** Thank you for including these works. I additionally agree with the literature references pointed out by Reviewer wz35 and would recommend the authors discuss these in the revision.

---

### Official Review · Reviewer_ssUw · 2023-11-01

**Soundness:** 3 good
**Presentation:** 3 good
**Contribution:** 2 fair
**Rating:** 5
**Confidence:** 4

**Summary:**

The authors propose meta-learning an optimizer for RL. They show that RL is uniquely challenging to optimize for. They then propose multiple techniques to learn an optimizer for RL from scratch that can generalize from toy tasks to Brax.

**Strengths:**

Originality:

- This is a new problem setting that I have not seen before. It is clear that existing learned optimizers do not perform well in RL and it is good to see initial works in this direction.

- The gradient processing is well-motivated and significant.

- The architecture is also well-motivated and elegant.

Quality:

- The authors perform neat investigations into hypotheses about gradient-related challenges in RL.

- The authors show impressive transfer performance.

Clarity:

- The paper is very clearly written.

Significance:

- This could ultimately lead to a superior optimizer for RL, which would be very significant.

**Weaknesses:**

Originality:

- There is a section missing from the related works. In particular, it's the "learning update rules / algorithms" for RL literature. The setup seems to be *very closely related to* the setup from "Discovering Reinforcement Learning Algorithms" (LPG) [1], which is part of a broader field of meta-learning general RL update rules [2], [3].

- The pipeline training and the training setup of LPG seem closely related.

Clarity:

- (Minor) Section 4.1: SeeAppendix B <= missing a space.

- See clarification-related questions below.

Quality:

- On Section 4.1: The authors show that the agent-gradient distribution is non-IID. However, they do not show that the gradient distribution for normal supervised learning (SL) **is** IID. This is rather important to show, if the authors are claiming that RL is a uniquely challenging setting.

- On Section 4.2: Again, the authors did not compare RL and SL, which is the purpose of this section. The authors could train a SL model and then see if the RNN can learn the identity function on the gradients from that training process.

- The synthetic data is not representative of the SL training process. Furthermore, the injection of non-iid dynamics of the synthetic data seems to have been done ad-hoc and is not particularly meaningful. For example, what if the iid shift was far more extreme?

- On Section 4.3: The problem of non-stationary targets is a well-studied phenomenon in RL, with plenty of possible prior works the authors could cite. This includes the deadly triad [4] and capacity loss [5].

Significance:

- The primary technical contributions seem to be the gradient processing, and the optimizer structure. The gradient processing is an impactful trick. The optimizer structure is hardly ablated or compared when there is plenty of literature on learned optimizer architectures.

- The authors show limited transfer and the optimizer does not seem to generally perform on-par with Adam, despite being heavily inductively biased towards an Adam-like update.


[1] Oh, Junhyuk, et al. "Discovering reinforcement learning algorithms." Advances in Neural Information Processing Systems 33 (2020): 1060-1070.

[2] Kirsch, Louis, Sjoerd van Steenkiste, and Jürgen Schmidhuber. "Improving generalization in meta reinforcement learning using learned objectives." arXiv preprint arXiv:1910.04098 (2019).

[3] Lu, Chris, et al. "Discovered policy optimisation." Advances in Neural Information Processing Systems 35 (2022): 16455-16468.

[4] Van Hasselt, Hado, et al. "Deep reinforcement learning and the deadly triad." arXiv preprint arXiv:1812.02648 (2018).

[5] Lyle, Clare, Mark Rowland, and Will Dabney. "Understanding and preventing capacity loss in reinforcement learning." arXiv preprint arXiv:2204.09560 (2022).

**Questions:**

1. What is the difference between your pipeline training and the pipeline training of LPG?

2. Is pipeline training desirable? Ideally we want a non-myopic optimizer, and the training dynamics early on in training heavily affect the distribution of parameters at the end of training. Can you ablate this?

3. Why is the neural network so small? Is this common in the literature?

4. In Section 6.1: Are you re-training STAR and L2LSGD, or are you taking pre-trained weights?

5. Why are the optimizers and environments different in each plot in Figure 4? (e.g. why is VeLO exclusively for Ant and STAR for big_dense_long). Can you generate a more complete plot here?

6. Many of the learned optimizers use ES to train their learned optimizers. Is there any particular reason you decided not to do this?

7. On Section 4.2: How did the authors choose the hyperparameters (the rate and total amount of change) for generating non-iid data?

8. On "Quality" Weaknesses (4.1 and 4.2) from above: These seem easy for the authors to address and I would be very curious about the results!

9. Is there a reason you did not try other architectures from the learned optimizer literature?

---

> ### Author Response · Authors · 2023-11-17
> **Reply to Reviewer ssUw**
>
> Thank you for your detailed feedback, suggestions, and comments. We address your points in the following.
> - **Add more ablation studies.** Please check the general response.
> - **The authors show that the agent-gradient distribution is non-IID in RL. However, they do not show that the gradient distribution for normal supervised learning (SL) is IID.** We did not claim that non-iid gradient distribution is a unique challenge for learning to optimize for RL. In fact, the gradient distribution in supervised learning is also non-iid. For example, there is a larger portion of zero gradients at the end of the training for both RL and SL. However, we expect the gradient distribution to be more iid in RL tasks since RL tasks are inherently more non-stationary. We will rewrite relevant parts to clarify this.
> - **The difference between pipeline training and the training style of LPG.** The major difference between pipeline training and the training style of LPG is that pipeline training is designed to make the agent-gradient distribution more iid. Pipeline training resets training units in a specific way such that the input agent-gradients are more diverse, spreading across a whole training interval.
> - **Why is the neural network so small?** We choose a small network so that we can train an optimizer within the computation and memory budget. This also helps reduce the deployment computation burden, i.e. reducing the inference computation of a fixed learned optimizer.
> - **Many of the learned optimizers use ES to train their learned optimizers. Is there any particular reason you decided not to do this?** Learned optimizers are mainly trained with the meta-gradient method and evolutionary method. Following previous works, we choose one of the methods to train our learned optimizer, i.e. the meta-gradient method. Using the evolutionary method is an interesting future direction.
> - **Add more citations and correct typos.** We will add those citations as well as fixing the typos.
> - **Are you retraining STAR and L2LGD2?** Yes, we retrain them from scratch.

---

> > ### Comment · Reviewer_ssUw · 2023-11-18
> > **Thank you for the reply**
> >
> > Thank you for the response and clarifying points.
> >
> > > more ablation studies
> >
> > These are very helpful. Thank you for including them. It would be great to see the updated manuscript before the end of the discussion period, if the authors could upload it.
> >
> > > We did not claim that non-iid gradient distribution is a unique challenge for learning to optimize for RL...the gradient distribution in supervised learning is also non-iid
> >
> > I completely agree with the fact that the gradient distribution in SL is non-iid (which is why I asked the question). The author's first sentence, that "they did not claim that non-iid gradient distribution is a unique challenge for learning to optimize for RL" is odd, given that Section 4 begins with:
> >
> > "Learned optimizers for SL are infamously hard to train, suffering from high training instability. Learning an optimizer for RL is even harder. In the following, we identify three issues in learning to optimize for RL."
> >
> > And then jump into section 4.1, which is titled "THE AGENT-GRADIENT DISTRIBUTION IS NON-IID".
> >
> > If the gradient distribution is generally non-iid in supervised learning as well, the point in Section 4.1 seems completely trivial.
> > The authors mentioned in the reply that they would "expect the gradient distribution to be more iid in RL tasks since RL tasks are inherently more non-stationary". I *completely agree with this*, which is *why I asked them to run the experiments showing this*. The point of the section should be to show that RL is *more nonstationary* than SL, but instead just shows that it is non-stationary (which is completely trivial and true in simple SL). The same applies to Section 4.2. This is a significant portion and point in the paper.
> >
> > Keep in mind that these are not difficult experiments to run and would greatly strengthen the claims in the paper.

---

> > > ### Author Response · Authors · 2023-11-20
> > > **Update about gradient distribution experiments**
> > >
> > > Thank you for your reply. We just got some new results about gradient distribution experiments.
> > >
> > > We train a neural network in the MNIST task for 10 epochs and collect gradients during training. The network is the same as the actor network used in training A2C in big_dense_long, except for the output layer. **As we expected, the gradient distribution is non-iid but it is more iid than the agent-gradient distribution in RL tasks.** To show this, we compute the Wasserstein distance (WD) between the (agent-)gradient distribution at different training stages and the distribution of all (agent-)gradients during training in the following table. Note that a smaller distance indicates a higher iid degree.
> > >
> > > |   task \ WD value  | WD(beginning, all) | WD(middle, all) | WD(end, all) |
> > > | -------------------- | ------------------------------ | ----------------------------- | ------------------------------ |
> > > |         MNIST          |  $4.0642 \times 10^{-4}$   |  $0.7205 \times 10^{-4}$  |  $1.4204 \times 10^{-4}$   |
> > > |  big_dense_long   |  $7.1583 \times 10^{-4}$   |  $1.1726 \times 10^{-4}$  |  $6.7967 \times 10^{-4}$   |
> > >
> > > We also approximated an identity function with gradients in MNIST as inputs and got similar results as in Figure 2. Specifically, without gradient processing, the accuracy is only about 30%; with gradient processing, the accuracy is much higher, around 80%. **As we claimed previously that non-iid gradient distribution is not a unique challenge for learning to optimize for RL; and we will modify related paragraphs and section titles to clarify this.** However, we kindly disagree with the claim that "the point in Section 4.1 is trivial." **As far as we know, the non-iid issue in learning to optimize is not explicitly identified in previous works, even though this issue could make training more unstable and reduce learning performance significantly. In our work, we 1) clearly point out this problem and 2) propose an effective method to mitigate it.**

---

> ### Comment · Reviewer_ssUw · 2023-11-21
> **Thank you for the additional results**
>
> Thank you for the additional results! Using the WD to the total distribution is a neat way to calculate the iid-ness. It would be good to see these plots and histograms. Are you able to update the paper with them?
>
> Minor nits: It would be good to make sure the problems are similarly-scaled. E.g. would it be possible to do something like doing behavioral cloning or contextual bandits on transitions sampled from an expert. This way, you assure that the input spaces match and the loss functions / objectives are effectively the same, but the problem is now iid.
>
> > the non-iid issue in learning to optimize is not explicitly identified in previous works
>
> This doesn't make it non-trivial. My understanding is that it is so commonly understood that it's not something most people would write in a paper. Adding the comparisons between RL and SL make it non-trivial. It's good to see the authors do this.
>
> I hope that the authors will update the paper so that we can see the latest results (that should hopefully replace Sections 4.1 and 4.2) in more detail as soon as possible. My understanding is that the authors can re-write and update the manuscript during the discussion period.

---

### Author Response · Authors · 2023-11-17
**General Response**

We thank all reviewers for the detailed feedback, suggestions, and comments. Many reviewers ask for ablation studies. Due to the high computation requirements of these tasks, we are still working on the ablation studies and we may not finish all experiments before the rebuttal deadline. Even so, we decided to provide a general response by presenting currently available results here. **In short, our current results show that both pipeline training and the inductive bias of the optimizer structure are essential, while gradient processing is less helpful.**

- **Ablation study of pipeline training.** As shown in the following table, without pipeline training, Optim4RL performs similarly to the full version in two simple gridworlds. However, Optim4RL without pipeline training performs much worse than the full version in more complex Brax tasks. Besides, pipeline training also makes Optim4RL more robust to hyper-parameters.

|           Method \ Task          | small_dense_short |   big_dense_long  |    Ant        |  Humanoid |
| ----------------------------------- | ------------------------- | ------------------------ | ------------- | -------------- |
|  with pipeline training | 11.31 $\pm$ 0.07  | 23.14 $\pm$ 0.33 |  7,088 $\pm$ 178  | 7,658 $\pm$ 282 |
| w.o. pipeline training | 11.15 $\pm$ 0.10  | 23.04 $\pm$ 0.41 |  5,603 $\pm$ 631 | 7,053 $\pm$ 272 |

- **Ablation study of the inductive bias of the optimizer structure.** The ablation study of the optimizer structure is already included in Section 6.1 (Figure 4). We will highlight this to make it clearer. Specifically, we compare Optim4RL with different optimizer structures and other SOTA optimizers, including LinearOptim, L2LGD2, STAR, and VeLO. Specifically, for LinearOptim, except for using a “linear” parameter update, the other components are the same as Optim4RL. So the comparison between Optim4RL and LinearOptim is an ablation study of the inductive bias of the optimizer structure. The results in Figure 4 show that LinearOptim fails to optimize simple RL tasks, proving that the inductive bias of Optim4RL is better. Some reviewers also suggest adding STAR to Figure 4(c). As we show in Figure 4, STAR failed in simple gridworlds, such as small_dense_short and big_dense_long (Figure 4(a,b)). So we expected that STAR was unlikely to work well in more complex tasks (e.g. Ant in Brax). For completeness, we are working on testing STAR in Ant and will share the result once we get it. For VeLO, we find it hard to train it within the computation budget. Instead, the performance of VeLO in Ant is taken from the VeLO paper.

- **Ablation study of gradient processing.** We did experiments and found that with or without gradient processing, Optim4RL achieves a similar best performance after hyper-parameter tuning. However, removing gradient preprocessing makes Optim4RL more robust to hyper-parameters. Given this result, we decide to remove gradient processing in our method and rerun all experiments. However, we do believe the results in Section 4.2 are interesting to report. We will move this section to the appendix and leave gradient representations for future investigation.

Thank you for helping improve our work.

---

> ### Author Response · Authors · 2023-11-20
> **Quick update about STAR experiments in Ant**
>
> We have finished experiments of meta-training STAR in Ant for Figure 4(c). The performance of STAR in Ant is quite unstable, achieving an average return of 1000 while our method Optim4RL can reach an average return of 7000.
>
> We will also update the draft to incorporate the required modifications by all reviewers with the new results.

---

> ### Author Response · Authors · 2023-11-22
> **Update Draft**
>
> Dear Reviewers,
>
> We have updated our draft to incorporate modifications by all reviewers with the new results. Specifically, we
> - add section 3.2 to discuss related works about discovering general reinforcement learning algorithms;
> - add ablation studies of pipeline training and the inductive bias of our optimizer Optim4RL in section 6.1 (We didn't include gradient processing ablation since this component may be moved to the appendix.);
> - add STAR and LinearOptim results in Figure 4(c). Note that LinearOptim in Ant fails due to NaN errors, probably due to highly non-stationary parameter updates;
> - add a discussion about m and n in section 5.1;
> - add more details about computing $\Delta \theta$ in section 5.3;
> - plot the agent-gradient distribution of all six gridworlds in Figure 6 as well as adding the related discussion;
> - update section 4.1 and 4.2 to discuss supervised learning setting and include related experiments in appendix E;
> - add more training details in appendix C;
> - include other minor changes, such as fixing typos and misleading claims.
>
> We thank all reviewers for providing your valuable suggestions. We are grateful for the opportunity to address your concerns and would like to express our sincere appreciation for considering a reevaluation of our work. In case we forget to include any important changes, please let us know.

---

### Meta-Review · Area_Chair_DPrx · 2023-12-07

**Metareview:**

In this paper, the authors analyse the dynamics of gradient updates in RL and identify three different problems (logarithmic range, non iid, variance). The authors use these insights to design a highly structured meta-learning space in which they meta-learn an optimiser.

One of the key strengths of the paper is that there has not been much work in the area of learned optimisers for RL and that learned optimisers from supervised learning do not transfer well to the RL problem.

However, there are also a few key weaknesses with the paper that make it a borderline paper (leaning reject) in my assessment:
First of all, the in-distribution performance of Optim4RL is dominated by ADAM, even though a lot prior structure (similar to Adam) is built into the meta-learner.
Secondly, the transfer performance of the algorithm is not on par with off the shelf optimisers such as Adam and sometimes fails quite badly. Again, this would be one thing if the optimiser was learned black-box without much structure, but is disconcerting when a lot of prior structure is baked into it.
Thirdly, the authors missed closely related work, such as LPG, and related literature in the area of "dormant neurons" in RL which provides a different explanation and remedy for some of the phenomena they observe.

Last (not least), one of the ablations requested by the reviewers revealed that an entire part of the method and narrative of the paper ("gradient pre-processing") did not actually improve the performance of the method.
Like the reviewer, I recommend the authors for running the experiments and being candid, but updating the paper and narrative based on these new results is a substantial change which warrants another full set of reviews.

**Justification For Why Not Higher Score:**

See "weakness" of my meta-review listed above.

**Justification For Why Not Lower Score:**

N/A

---

### Decision · Program_Chairs · 2024-01-16

Reject